# Study on Erosion Behavior of Laser Wire Feeding Cladding High-Manganese Steel Coatings

**DOI:** 10.3390/ma16175733

**Published:** 2023-08-22

**Authors:** Huafeng Guo, Chenglin Zhang, Yibo He, Haifeng Yang, Enlan Zhao, Longhai Li, Shaohua He, Lei Liu

**Affiliations:** 1School of Mechanical and Electrical Engineering, Xuzhou University of Technology, Xuzhou 221018, China; starstory_z@163.com (C.Z.); lb21050034@cumt.edu.cn (E.Z.); longhaicumt@163.com (L.L.); 18061392689@163.com (S.H.); leiliu86@sina.cn (L.L.); 2State Key Laboratory of Solidification Processing, Center of Advanced Lubrication and Seal Materials, Northwestern Polytechnical University, Xi’an 710072, China; heyibo@nwpu.edu.cn (Y.H.); 4841@cumt.edu.cn (H.Y.); 3School of Mechatronic Engineering, China University of Mining and Technology, Xuzhou 221116, China

**Keywords:** laser wire feeding cladding, high-manganese steel coating, deformation hardening, erosion mechanism

## Abstract

High-manganese steel (HMnS) coating was prepared using laser wire feeding cladding technology. Erosion damage behavior and erosion rate of both the HMnS coating and the HMnS substrate were investigated at room temperature using an erosion testing machine. SEM/EDS, XRD, EPMA, and microhardness analyses were used to characterize the cross sections of the coating and matrix, as well as the morphology, phase composition, and microhardness of the eroded surface. The phase composition, orientation characteristics, and grain size of the eroded surfaces of both the coating and substrate were examined by using the EBSD technique. The erosion mechanism under different erosion angles was revealed. By analyzing the plastic deformation behavior of the subsurface of the HMnS coating, the impact hardening mechanism of the high-manganese steel coating during the erosion process was investigated. The results demonstrated that the HMnS coating, prepared through laser wire feeding cladding, exhibited excellent metallurgical bonding with the substrate, featuring a dense microstructure without any cracks. The erosion rate of the coatings was lower than that of the substrate at different erosion angles, with the maximum erosion rate occurring at 35° and 50°. The damage to the coating and substrate under low-angle erosion was primarily attributed to the micro-cutting of erosion particles and a minor amount of hammering. At the 90° angle, the dominant factor was hammering. After erosion, the microhardness of both the coating and substrate sublayer increased to 380HV0.3 and 359HV0.3, respectively. Dendrite segregation, refined grains, low-angle grain boundaries, and localized dislocations, generated by laser wire feeding cladding, contributed to the deformation process of HMnS. These factors collectively enhance the hardening behavior of HMnS coatings, thereby providing excellent erosion resistance.

## 1. Introduction

Erosion refers to the phenomenon of material surface loss caused by the impact of fluid or solid particles at a specific speed and angle. Erosion is prevalent across various industrial sectors, such as machinery, metallurgy, energy, building materials, transportation, aviation, aerospace, and the military industry. It constitutes a common form of wear in modern industrial production and has evolved into a significant cause of equipment failure and material damage. The lifespan of aircraft engines flying in dusty areas, for instance, can be reduced to merely 1/10 of their normal expectancy. Approximately one-third of boiler pipelines accidents can be attributed to erosion wear. This type of wear primarily arises from micro-cutting and impact fatigue induced by high-speed particles striking the surface of components [1,2,3,4]. Consequently, studying the evolution of material surfaces and sub-surfaces under high-speed impact holds immense significance in enhancing the erosion resistance of components.

HMnS demonstrates commendable work-hardening properties and impact resistance. Its surface undergoes rapid work hardening, which improves surface hardness while maintaining internal toughness. Song et al. [5] investigated the microstructure evolution in ultra-high-manganese steel during heat treatment, unveiling the wear-hardening mechanism of this material. Zhang et al. [6] employed molecular dynamics simulations to elucidate the interaction mechanism between dislocations and coherent twin boundaries by studying the relationship between dislocation slip rate and shear rate during plastic deformation. Wang et al. [7] deposited an HMnS coating onto the Q235 surface, exploring the effects of heat treatment on surface hardness, wear resistance, and the depth of the hardened layer after wear.

The aforementioned research indicates that HMnS can achieve a surface hardness of 300–400 HB under low impact loads, and this hardness can increase to 500–800 HB under high impact loads. Yun et al. [8] from Hanyang University prepared as-cast austenitic Fe–Cr–C–Mn alloys with varying Mn contents in a vacuum induction melting furnace. They investigated the sliding wear characteristics and the influencing mechanisms in the hardened layer formed through structural transformation in different strain-induced martensite phases. Consequently, HMnS is frequently utilized for manufacturing excavator shovel teeth, rolling and crushing walls for conical crushers, fork plates for jaw crushers, lining plates for ball mills, railway switches, plate hammers, hammer heads, and more.

Currently, research on work-hardening in high-manganese steel primarily emphasizes strong impact wear [9,10] and friction wear [11,12], with comparatively fewer studies examining erosion wear of high-manganese steel due to solid particles. Yuping et al. [13] employed HVOF technology to create a Fe–Cr–Si–B–Mn coating and disclosed its cavitation erosion resistance. The coating exhibited superior resistance to cavitation erosion compared to the original substrate steel due to its higher hardness. Kim et al. [14] investigated the comparative erosion resistance of Fe–Cr–C–Ni/Mn austenitic alloys with varying Fe–Cr–C–Ni/Mn content, unveiling the influence of strain-induced martensite on erosion performance and mechanisms. The findings indicated a phase transition from austenite to martensite under distinct erosion conditions. Simultaneously, at a 30° erosion angle, the material primarily undergoes ploughing and shear lip damage, while at 90°, numerous erosion pits and extrusion lips emerged on the damaged material surface. Yildizli et al. [15] employed a high-manganese steel electrode to create a high-manganese steel surfacing layer on low-carbon steel. They investigated the microstructure, hardness, and characteristics of solid particle erosion in the high-manganese steel surfacing layer. The results indicated that the erosion rate varied with the number of surfacing layers and the erosion angle, with the highest erosion rate across various process samples occurring at 90°.

Wang et al. [16] examined the impact abrasive wear characteristics and mechanisms in ultra-high-manganese steels Mn18 and Mn13, both at high and low speeds over extended durations. TEM and other methods revealed that the sub-impact layer primarily comprised high-density dislocations and slip bands, contributing to dislocation strengthening. Additionally, under high-impact conditions, the hardness of Mn18 rapidly increased to 440 HB. 

Wang et al. [17] conducted a comparative study using a hot-rolled medium manganese steel plate and an NM450 martensitic wear-resistant steel plate. Their study mainly uncovered the erosion and hardening mechanisms of the two materials subjected to sediment erosion wear. The findings demonstrated that the wear-hardening mechanism primarily resulted from a combination of twin strengthening, dislocations, and strain-induced martensitic transformation. The erosion mechanism was mainly manifested by micro-cutting, material spalling due to plastic deformation, and localized chiseling.

Due to its excellent work-hardening performance, HMnS is challenging to machine in practical production, with most of its applications focused on casting parts [18]. However, the high manganese content leads to significant component segregation and inclusion formation, resulting in thermal cracking during solidification and impact cracking during usage [18]. Moreover, severe working conditions such as strong impacts and heavy wear can induce cracks due to casting defects or stress concentrations [19,20]. 

Wear-resistant HMnS boasts a high carbon content (ranging from 1.0 to 1.4%), low thermal conductivity (approximately one-third that of carbon steel), a high linear expansion coefficient (roughly twice that of carbon steel), and poor weldability. It was once deemed a non-weldable material. Jong et al. [21,22] delved into compositional segregation in gas tungsten arc welded (GTAW) Fe24.2Mn3.4Cr0.44C austenitic HMnS, revealing the mechanism by which GTAW welding affects strain hardening rate and strain rate sensitivity. Currently, the repair/remanufacturing of HMnS often involves using a small current, low speed, intermittent welding method. Water cooling and intermittent hammering are required during the process to minimize carbide precipitation and residual tensile stress. However, this approach results in lower welding efficiency and poorer welding quality [19,23].

While HMnS boasts excellent comprehensive mechanical properties, its widespread application in other engineering fields has been impeded by significant component segregation and inclusion formation during casting, compromised hardening performance during mechanical processing, and limited formability during high-temperature processing. Due to the characteristics of small heat input and rapid cooling during laser welding, laser welding technology offers a new avenue for the application of HMnS [18]. In this paper, a laser wire feeding cladding system was employed to fabricate HMnS coatings, with a focus on exploring their erosion wear performance. Through an investigation into the microstructure of the erosion surface and sub-surface, we aim to reveal its erosion resistance. 

## 2. Experiments and Methods

### 2.1. Experimentation

The laser wire feeding cladding system comprised a laser generation system, a three-dimensional platform, a wire feeding machine, a chiller, and a protective gas system, as illustrated in Figure 1a. The laser generation system primarily featured a continuous fiber laser equipped with a coaxial gas nozzle (HWL-R1500W, Dongguan Huawei laser equipment Co., Ltd., Dongguan, China). The fiber laser operated at a wavelength of 1064 nm and delivered a power of 1500 W. 

During the laser wire feeding cladding process, the 3D platform followed a scanning path determined by the computer control system. To prevent metal oxidation and pore formation by isolating oxygen from the air, a protective gas was essential. In this regard, 99.99% high-purity argon gas was employed as the protective gas. Throughout the experiment, wire feeding was executed in a forward position, as shown in Figure 1b. Referring to the wire feeding parameters used in laser wire filling welding [24,25], the laser wire feeding cladding process for HMnS coatings was conducted with the parameters detailed in Table 1.

In the experiment, a Mn13 plate (manufactured by Nine Steel Co., Ltd., Shanghai, China) created through the continuous casting method was selected as the substrate. Its composition is detailed in Table 2. The composition of the welding wire (Tianjin Bridge Welding Materials Group Co., Ltd., Tianjin, China) used for laser wire feeding cladding is also presented in Table 2. The HMnS coating, produced through laser wire feeding cladding, was manufactured into two sizes of test samples using electric spark wire cutting. Samples sized at 29 mm × 29 mm × 10 mm were designated for erosion wear testing, while 10 mm × 10 mm × 10 mm samples were utilized for hardness testing and microstructure characterization. Both sets of samples were cleaned and dried with alcohol before and after the erosion tests. The mass loss was quantified using an electronic balance with an accuracy of 0.1 mg. For microstructure observation and hardness testing, the samples were ground with silicon carbide sandpapers and finally polished on a metallographic grinder. Before observing the microstructure of the HMnS section, the sample’s surface underwent corrosion using a solution of HNO_3_: HCl in a 1:3 ratio, forming aqua regia.

### 2.2. Testing Methods

The cross-sectional microstructure and surface morphology of HMnS were characterized using a metallographic microscope (OM, Leica DM4, Wetzlar, Germany) and a field emission scanning electron microscope (SEM, TESCAN MIRA, Brno, Czech Republic). The composition of the HMnS surface after erosion was analyzed with an energy spectrum analyzer (EDS, QUANTAX Bruker, Karlsruhe, Germany) attached to the SEM. The phase composition of both the HMnS plate and the HMnS coating was analyzed using a diffractometer (Bruker D8 Advance, Karlsruhe, Germany). 

For the observation of elemental distribution in the coating cross-section, a field emission electron probe microanalyzer (EPMA 8050G, SHIMADZU, Kyoto, Japan) was employed. The hardness changes of both the HMnS plates and HMnS coatings before and after erosion were detected using a micro hardness tester (HVT-1000A, Yantai, China) with 300 g loads and 15 s hold time. Furthermore, the microstructure and grain size of the samples were characterized via electron backscatter diffraction (EBSD, QUANTAX Bruker, Karlsruhe, Germany). The erosion experiments were conducted using a self-designed room temperature erosion testing machine. The working principles of the erosion testing machine are shown in Figure 2. The erosion test mainly studied the relationship between the erosion rate of the HMnS plate and HMnS coating, considering changes in the erosion angle. An analytical balance with an accuracy of 0.1 mg was employed for weighing, and the average value was derived from three consecutive weighings. Alumina particles with an average particle size of 60 mesh were used as erosion material. The erosion parameters were as follows: erosion pressure of 0.8 MPa, erosion time of 20 s, erosion distance of 90 mm, sand flushing amount of 150 g, and erosion angles of 20°, 30°, 35°, 40°, 50°, 60°, 70°, 80°, and 90°, respectively. Formula (1) was utilized for calculating the erosion rate.
(1)$$\sigma =\frac{\bar{m_{1}}-\bar{m_{2}}}{M}$$

In the equation, σ represents the erosion rate (mg/g), $\bar{m_{1}}$ is mass before erosion (mg), $\bar{m_{2}}$ is the mass after erosion (mg), and *M* represents the mass of the erosion sand quality (g).

## 3. Results and Discussions

### 3.1. Microstructures of HMnS Coating

The microstructure of the laser wire-fed cladding HMnS coating significantly influences its erosion performance. Figure 3 displays the optical microscopy (OM) image and X-ray diffraction (XRD) spectrum of the cross-sectional microstructure. In Figure 3a, the yellow curve indicates the overlapping interface of two cladding lines. The grain morphology differs on either side of the interface. The cladding line on the left side overlaps the cladding line on the right side. During the solidification process of the right cladding line, distinct grain shapes emerged due to varying temperature gradients and heat dissipation directions in the middle and top regions. Below the blue dashed line, columnar grains are visible, while above it, equiaxed dendritic grains are present. As the cladding line on the left side of the interface solidified, the overlapping area epitaxially grew onto the right cladding line, acting as the substrate and forming columnar grains below the blue dashed line. As the temperature gradient decreased and the heat dissipation direction changed, equiaxed dendritic grains eventually developed at the top. Figure 3a illustrates that the solidification structures exhibit relative density, devoid of evident solidification defects and microcracks.

Figure 3b presents the XRD spectrum. The diffraction peaks indicate primarily austenitic structures, with a minimal presence of ferrite. Therefore, this paper does not discuss excessive carbide precipitation within the cladding layer. The excessive precipitation of carbides significantly impacts the performance of HMnS during casting and arc welding processes. The HMnS coating produced through laser wire feeding cladding exhibits substantial differences from HMnS produced using traditional casting and arc welding methods. During casting, as the temperature cools between 950 °C and 300 °C, carbon swiftly precipitates at the austenite grain boundaries, resulting in the formation of network carbides. In comparison, arc welding entails higher heat input, resulting in more carbide precipitation in arc-welded HMnS [13]. Furthermore, instances of penetrating cracks and microcracks occasionally accompany casted HMnS and arc-welded HMnS. 

Being a heat source with concentrated energy, the rapid cooling rate during the laser remelting of high-carbon HMnS prevents carbide precipitation in the solidified microstructure [26,27]. Consequently, similar to the laser remelting process for high-carbon HMnS, the low heat input and rapid cooling rate in laser wire feeding cladding prevent carbon from precipitating at the austenite grain boundaries, thereby inhibiting carbide formation along the grain boundaries. 

Figure 4 depicts the SEM images of the cross-sectional microstructure of the HMnS coating. In Figure 4a, the lower microstructure is visible, comprising both the solidified structure and the substrate. The dashed line demarcates the interface between the solidified structure and the substrate. Beneath this interface lies a single austenite structure with coarse grains. Laser wire feeding cladding constitutes a rapid melting-solidification hot working process, owing to its concentrated heat, minimal heat input, and small heating area. Given the low thermal conductivity of HMnS, just one-third that of carbon steel, the temperature gradient at the solidification interface becomes significant, causing the solidification structure to grow upwards in the form of planar grains. This is illustrated in Figure 4a at the interface. As the solidification rate increases, solute diffusion leads to the instability of the internal structure, causing the growth mode of grains to shift from planar to columnar. This transformation is evident in the region above the interface. As growth continues, the heat within the molten pool rapidly transfers downwards through the HMnS substrate, resulting in a perpendicular columnar grain structure at the interface. Furthermore, longer columnar grains form in the central area of the molten pool, as depicted in Figure 4b. During the gradual solidification of the molten pool, heat conduction occurs not only towards the solidified area but also into the surrounding air. According to the component undercooling theory, the high degree of undercooling leads to the formation of a typical equiaxed dendritic grain structure at the upper part of the melt pool (Figure 4c). 

In summary, the microstructure of the HMnS coating created through laser wire feeding cladding aligns with that of laser-remelted HMnS [11,26]. Across the span from the interface to the upper part of the melt pool, distinct grain structures emerge, such as planar grains, columnar grains, and equiaxed dendritic grains without the precipitation of carbides.

### 3.2. Erosion Performance of HMnS

Figure 5 illustrates the variation in erosion rates with respect to erosion angles. To compare and analyze the erosion resistance of HMnS plates and HMnS coatings, erosion rates were examined under different erosion angles. It can be observed that while keeping the erosion distance, erosion time, pressure, and sand flushing amount constant, an increase in the erosion angle leads to an initial rise followed by a subsequent decline in the erosion rates for both plates and coatings. Notably, the erosion rates of the HMnS coatings consistently remain lower than those of the HMnS plates.

Figure 5 reveals that the highest erosion rate of HMnS plates occurs at an erosion angle of 35°, reaching a peak value of 1.0700 mg/g. As the erosion angle increases further, the erosion rate gradually diminishes. The lowest erosion rate is observed at an erosion angle of 90°, measuring 0.5860 mg/g, which is only 54.8% of the rate at 35°. For HMnS coatings, the maximum erosion rate is recorded at 50°, registering at 0.8378 mg/g. Similarly, the minimum erosion rate occurs at 90°, with a value of 0.5740 mg/g, representing 68.5% of the rate at 50°. These findings highlight that both HMnS plates and coatings exhibit notable work hardening effects during vertical impact, consequently resulting in reduced erosion rates.

Figure 6 presents the XRD spectrum following 90° erosion. After erosion, a small amount of Al_2_O_3_ particles becomes embedded in both the HMnS plate and the HMnS coating, concurrently inducing a minor amount of martensite. However, the austenite phase still dominates their surfaces. By comparing the XRD spectrum before and after erosion, it becomes evident that the erosion process subjects the surfaces of HMnS to substantial impact forces. This, in turn, leads to significant plastic deformation on surfaces, extensive grain refinement, and the emergence of notable residual stresses. Collectively, these phenomena contribute to the attenuation, displacement, and broadening of the diffraction peaks.

In Figure 7a, the microstructure and element distribution of the HMnS plates’ erosion surface at a 35°erosion angle are depicted. At this relatively small erosion angle, the surface displays numerous cutting marks and a few impact pits. Evidently, the erosion process is dominated by a cutting effect, resulting in greater material loss. 

Moving on to Figure 7b, the microstructure and element distribution of the erosion surface at a 90° erosion angle are presented. At high erosion angles, the surface undergoes pronounced plastic deformation due to high-speed impacts, leading to a substantial number of plastic deformation pits. The energy spectrum results in Figure 7 show the presence of a certain amount of Al element on the erosion surface. This suggests that some Al_2_O_3_ particles were fractured during the erosion process and subsequently embedded into the HMnS plate surface due to the high-speed impacts.

In Figure 8a, the microstructure and element distribution of the HMnS coating’s erosion surface at a 50°erosion angle are presented. Resembling the surface morphology of the HMnS plate subjected to 35° erosion, the HMnS coating surface exhibits numerous impact pits and cutting marks. The combined effects of micro-cutting and impact result in the creation of chips and significant material loss. Concurrently, a portion of Al_2_O_3_ particles remains on the surface. Shifting attention to Figure 8b, the microstructure and element distribution of the erosion surface at a 90°erosion angle are shown. Notably, the surface primarily displays impact pits caused by high-speed impacts at this angle. The energy spectrum results indicate the presence of a certain amount of Al_2_O_3_ particles on the erosion surface.

### 3.3. Erosion Mechanism

#### 3.3.1. Erosion Deformation of HMnS Surface

The sample exhibits more intricate erosion damage morphology due to the high velocity and random impact of numerous polygonal Al_2_O_3_ particles. This complexity arises from factors such as hardness, plastic toughness, internal defects, and surface distribution, which collectively influence the extent of erosion damage [27]. Figure 9a and Figure 9b illustrate the erosion surface morphology of the HMnS plate and the coating, respectively, at the angle corresponding to the maximum erosion rate. 

Figure 9a reveals that the damage characteristics of the matrix are primarily the result of narrow and elongated furrows caused by the ploughing action of erosion particles. Concurrently, significant plastic deformation occurs, leading to the formation of shear lips. Furthermore, the continuous impact of subsequent particles contributes to the fatigue spalling of these shear lips. The randomness of particle impacts results in furrows lacking a distinct directional pattern. 

Turning to Figure 9b, the erosion surface displays shorter furrows and a discernable quantity of impact craters. This outcome corresponds to the erosion angle of 50°, associated with the maximum erosion rate. At this angle, the vertical component force exerted by the erosion particles increases, while the influence of the tangential component force decreases. Consequently, the material surface experiences erosion damage due to both impact and ploughing.

As depicted in Figure 9c,d, the damage morphology of both the coating and the substrate during 90° erosion displays similarities, featuring numerous erosion pits, extrusion lips, and varying degrees of brittle spalling. This similarity arises from the predominant vertical impact of solid particles on the sample’s surface. 

Initially, a considerable number of erosion pits and extrusion lips form. With the successive impacts of particles, a work hardening effect is induced. This increase in material brittleness, coupled with heightened internal stress, leads to the generation of microcracks within the material’s hardening layer. Concurrently, the extrusion lips are prone to brittle fatigue spalling. If these microcracks within the erosion-hardened layer surface expand further, fatigue spalling occurs [15]. Consequently, the primary erosion mechanism for both the substrate and coating under 90° erosion comprises erosion pits and the occurrence of impact-induced fatigue spalling.

#### 3.3.2. Erosion Deformation of HMnS Sub-Surface

Deformation hardening is a pivotal characteristic of HMnS. When subjected to impact loads, HMnS undergoes deformation strengthening, leading to substantial enhancements in its strength, hardness, and wear resistance. Meanwhile, the internal austenite structure remains unchanged, maintaining the inherent plasticity and toughness of the material’s internal structure. Presently, the widely accepted mechanism behind the deformation hardening of HMnS is the twin strengthening theory. Upon impact, HMnS generates a multitude of twins that segment its matrix structure into numerous blocks. This process refines the grains, blocking dislocations, and hinders their propagation. Consequently, it becomes more challenging for HMnS to undergo further plastic deformation, thereby enhancing its deformation hardening capability [28]. However, compared to the formation of dislocations, the formation of twinning necessitates higher internal stress. Typically, twinning initiates and expands at grain boundaries or defects within the grains of HMnS.

Unlike the microstructure of cast HMnS, the microstructure of HMnS coating produced through laser wire feeding cladding impacts its strengthening mechanism. The grains in cast HMnS, following water toughening treatment, exhibit equiaxed grains with component segregation at grain boundaries and certain defects within the grains. During the deformation process, these grain boundaries and defects serve as nucleation sites for twins. This mechanism divides the grains by initiating and expanding twinning, which, in turn, impedes dislocation movement and achieves a strengthening effect. 

In contrast, the HMnS coating created via laser wire feeding cladding not only features grain boundaries similar to those produced from casting, but also develops smaller columnar and equiaxed dendrites within the grains. This occurs due to the substantial temperature gradient and component undercooling within the molten pool, as illustrated in Figure 3 and Figure 4. Adjacent columnar grains and equiaxed dendrites display a sight orientation difference, forming what is known as a low-angle grain boundary. In the process of laser wire feeding cladding, heat is concentrated, the melt pool size remains small, and solidification occurs rapidly. Consequently, the rapid solidification in the molten pool induces solute redistribution, leading to component segregation near low-angle grain boundaries, as depicted in Figure 10. Figure 10b,c illustrate that columnar grains contain a higher concentration of Fe elements, whereas Mn elements are more prevalent between these columnar grains. Christian et al. [18,29] employed the laser metal deposition method to fabricate low-carbon HMnS part and found that Mn elements also segregate within low-angle grain boundaries. They conducted a comprehensive study on the effect of component segregation on fault formation energy and the hardening behavior of low-carbon HMnS during tensile testing. Their research unveiled the mechanism behind twin formation and hardening within a heterogeneous structure. Therefore, the dendrite segregation generated by laser wire feeding cladding alters the dislocation and twinning formation during HMnS deformation process, consequently modifying the hardening behavior of the HMnS coating.

Using 90° erosion as an example, let us compare and analyze the influence of microstructure on erosion mechanisms between HMnS plates and HMnS coatings. The phase composition and grain size significantly impact the mechanical properties of materials. Figure 11 displays the phase composition and grain size statistics after erosion. The phase diagram (Figure 11a,b) indicates the predominant presence of equiaxed austenite grains with relatively uniform grain sizes. Due to the polishing treatment applied to the HMnS coating’s surface prior to erosion, the upper equiaxed dendrites have been removed. Consequently, HMnS coatings mostly consist of columnar grains aligned along the build direction, with a maximum grain length and width of approximately 400 μm and 150 μm, respectively. Smaller grains with diameters around 10 μm are also present. Figure 11c shows the statistical distribution of grain sizes farther away from the erosion layer. It is evident that the grain size of the HMnS plate is relatively concentrated, boasting an average size of 42.05 μm. On the other hand, HMnS coating predominantly features smaller grain sizes, with larger grains present in the overlap zone, resulting in an average grain size of merely 31.07 μm. These findings indicate that the rapid solidification principle employed during laser wire feeding cladding yields a comparatively smaller grain size for the HMnS coating. Consequently, the HMnS coating exhibits enhanced mechanical properties and improved erosion resistance.

Figure 12 displays the reverse pole diagram and pole diagram at a distance from the erosion layer. In the direction of erosion, the grains exhibited random orientation. The grain orientation distribution of the HMnS plate was relatively uniform, with a slight aggregation in the <111> direction. On the other hand, the grain orientation of the HMnS coating showed concentration primarily along the <001> direction near the erosion layer. 

This phenomenon can be attributed to the erosion of the surface layer by numerous high-speed particles, which induces significant slips and dislocations within the surface grains. The interaction, as illustrated in Figure 13, results in grain deflection and plastic deformation, ultimately altering the grain orientation. In regions far from the erosion layer, the grains were predominantly oriented along the <101> direction, with a minority of grains oriented along <111>. 

Examining the pole diagram in Figure 12, the pole density for the HMnS plate was 4.44, while for the HMnS coating, it was 8.41, indicating a strong preferred orientation. This preferred orientation of the HMnS coating can be attributed to its manufacturing process. During laser wire feeding cladding, the small size of the melt pool and the focused heat concentration cause high temperature gradients and specific heat flow directions. As a result, the HMnS coating experiences specific directionality and a preferred orientation in terms of grain growth. 

Dislocation strengthening constitutes the primary method for material strengthening. Figure 13 illustrates the Kernel Average Misorientation (KAM) and Grain Boundary (GB) diagrams. KAM calculation enables the analysis of variations in local dislocation density and local strain. The color gradient, transitioning from blue to red, signifies the orientation difference between adjacent grains. It can be calculated that the average KAM value of the HMnS plate, located far from the erosion layer, was 0.22, while the average KAM value of the erosion layer was 0.66. 

Referring to Figure 13b, it becomes evident that subgrain boundaries were nearly absent in regions far from the erosion layer. Conversely, within the erosion layer, a multitude of subgrain boundaries (<2°) emerged. The formation of subgrain boundaries mainly results from the interaction between dislocation slip and twinning, induced by plastic deformation on the surface of HMnS. This interaction leads to significant surface hardening in the HMnS plate. Figure 13c,d display the KAM and grain boundary diagrams of the HMnS coating. The average KAM value far from the erosion layer in the HMnS coating was 0.6, while the average KAM value within the erosion layer was 0.69. 

In general, the average KAM value of the HMnS coating was higher than that of the HMnS plate. The elevated average KAM value in the HMnS coating, far from the erosion layer, mainly originates from the presence of numerous low-angle grain boundaries (as depicted in Figure 13d) and component segregation due to rapid solidification (as shown in Figure 10). The heightened average KAM value within the erosion layer mainly arises from the interaction between the dislocation slip and twinning during impact-induced deformation. Consequently, the fine grain strengthening effect in the HMnS coating, combined with the reduction in mean free path hindering dislocation movement, results in superior mechanical properties of the HMnS coating compared to the HMnS plate.

The Schmidt factor of a material influences its strengthening mechanism. Figure 14 shows the Schmidt factor distribution of HMnS in the Y0 direction and the hardness distribution of the HMnS section after erosion. The results reveal that the average SF value and the minimum SF value far from the erosion area on the HMnS plate were 0.460 and 0.32, respectively. Similarly, for the HMnS coating, the average SF value and the minimum SF value far from the erosion area were 0.461 and 0.28, respectively. 

It can be seen that there is minimal difference in the Schmidt factor between the HMnS plate and the HMnS coating. However, the distribution of Schmidt factors in the HMnS plate appears relatively uniform, while the surface of the HMnS coating primarily exhibits high Schmidt factor values. This observation indicates the prevalence of soft-oriented grains in the Y0 direction of the HMnS coating. Therefore, during the erosion process, the presence of a high Schmidt factor on the surface layer of the HMnS coating facilitates easier grain slipping. This leads to an increased occurrence of dislocations and grain deformation, ultimately resulting in the formation of a deeper hardened layer. After undergoing 90° erosion, the hardness distribution of the cross sections of both the HMnS plate and the HMnS coating is depicted in Figure 14c. The hardened layer depth caused by erosion in the HMnS coating exceeded that of the HMnS plate. This increase in hardened layer depth can be attributed to the combined effects of component segregation, grain refinement, low-angle grain boundaries, and localized dislocations present in the HMnS coating. The cross-sectional hardness of the HMnS coating is generally higher than that of the HMnS plate, as depicted in Figure 14c. Therefore, throughout the erosion process, the HMnS coating demonstrates superior erosion resistance when compared to the HMnS plate.

In summary, at the angle associated with the highest erosion rate, the combined effect of tangential and forward forces leads to the most substantial mass loss. At elevated erosion angles, the forward force predominates, and Al_2_O_3_ particles continuously forge, compress, and crush the surface, resulting in the peeling of the HMnS surface. However, due to their work hardening characteristics, their erosion rates at high angles are relatively low. Moreover, the presence of severe component segregation, refined grains, numerous low-angle grain boundaries, and localized dislocations contributes to the enhanced hardening attributes of the HMnS coating. Therefore, the overall erosion resistance of the HMnS coating surpasses that of the HMnS plate.

## 4. Conclusions

Laser wire feeding cladding technology was employed to manufacture HMnS coatings. A self-designed erosion testing machine was utilized to assess the erosion resistance of the HMnS coating at various erosion angles, and an analysis of the erosion mechanism of the HMnS coating was conducted. The primary conclusions derived from this study are as follows:

The HMnS coating was successfully synthesized on the surface of the HMnS plate using laser wire feeding cladding technology. The coating comprised solely a single austenite phase and was characterized by a dense microstructure and the absence of noticeable defects. The microstructure of the HMnS coating demonstrated a transition from planar grains at the bottom to columnar grains and equiaxed dendrites as one progressed upwards through the coating. 

A self-designed erosion testing machine was utilized to conduct erosion tests on both HMnS plates and HMnS coatings at varying erosion angles. It was observed that, with an increase in the erosion angle, the erosion rate initially rose before subsequently declining. The angles corresponding to the maximum erosion rate for HMnS plates and HMnS coatings were determined to be 35° and 50°, respectively, while both reached their lowest erosion rate at an angle of 90°. The erosion rate of the HMnS coatings consistently demonstrated values lower than that of HMnS plates, indicating the superior erosion resistance of the coatings. Therefore, the overall erosion resistance of HMnS coatings surpassed that of HMnS plates.

At the angle associated with the highest erosion rate, the combined effect of tangential and forward forces resulted in the most significant mass loss. At high erosion angles, the forward force dominated, and the work hardening characteristics of HMnS led to a reduced erosion rate. The presence of severe component segregation, refined grains, numerous low-angle grain boundaries, and localized dislocations contributed to the enhanced hardening attributes of HMnS coatings.

## Figures and Tables

**Figure 1 materials-16-05733-f001:**
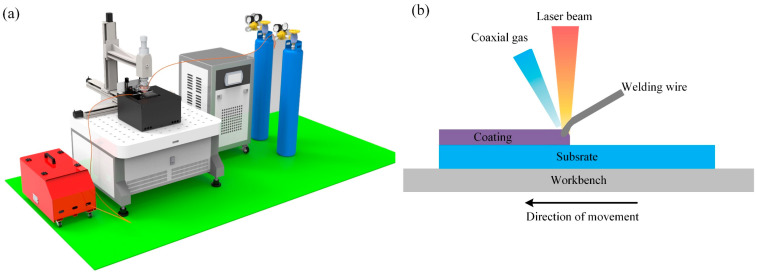
Principle of laser wire feeding cladding: (**a**) laser wire feeding cladding system, (**b**) Schematic diagram of laser wire feeding cladding principle.

**Figure 2 materials-16-05733-f002:**
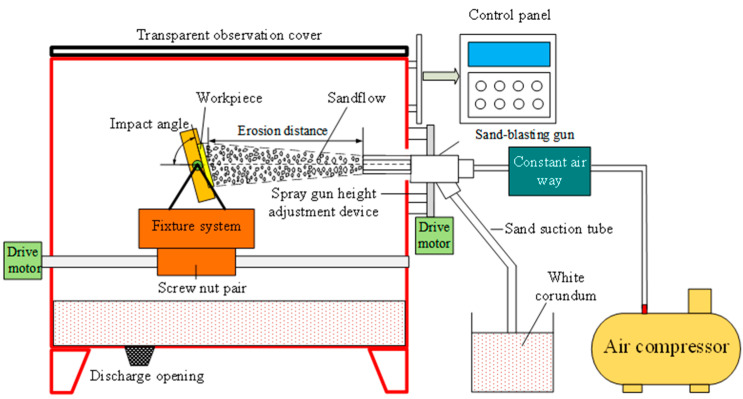
Schematic diagram of room temperature erosion test machine.

**Figure 3 materials-16-05733-f003:**
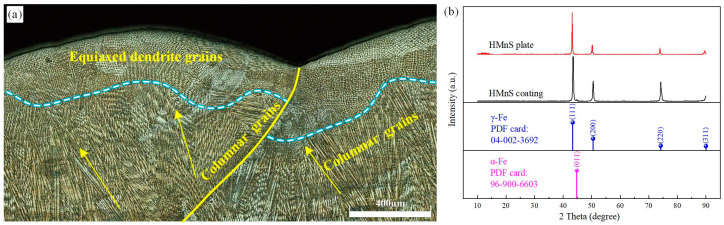
OM image and XRD spectrum; (**a**) Cross-section microstructure of HMnS coating; (**b**) XRD spectrum.

**Figure 4 materials-16-05733-f004:**
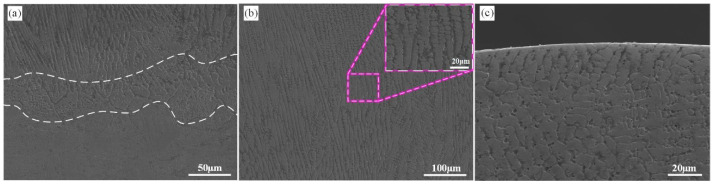
SEM images of the cross-section microstructures of laser wire feeding cladding HMnS coating; (**a**) bottom microstructure; (**b**) middle microstructure; (**c**) top microstructure.

**Figure 5 materials-16-05733-f005:**
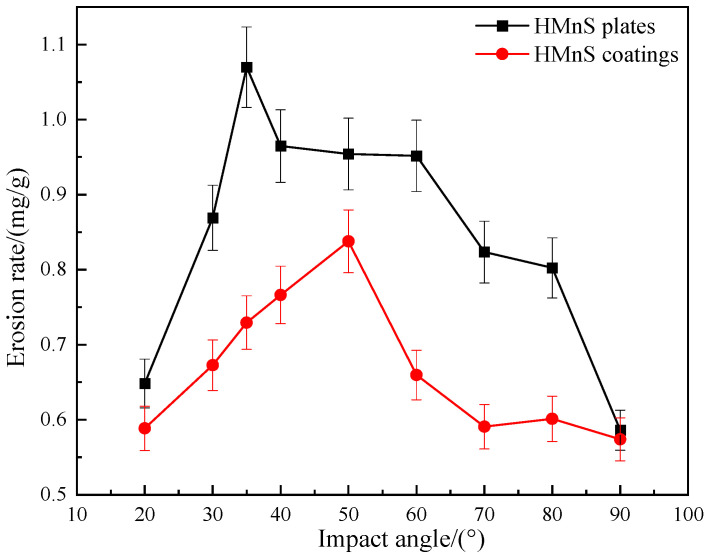
Variation of erosion rate with erosion angles.

**Figure 6 materials-16-05733-f006:**
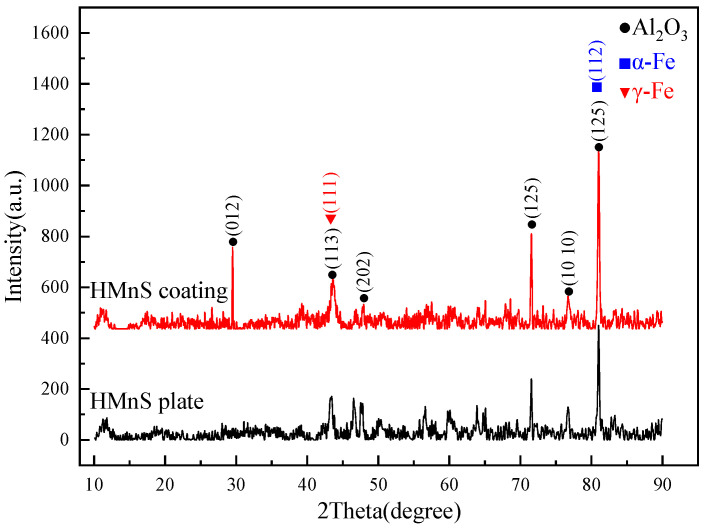
XRD spectrum after 90° erosion.

**Figure 7 materials-16-05733-f007:**
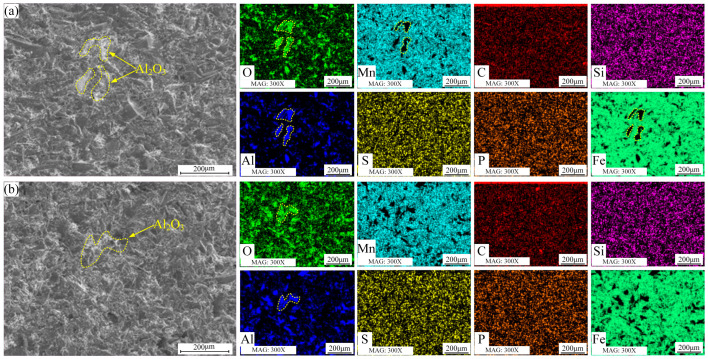
Surface morphology and element distribution of HMnS plate after erosion; (**a**) erosion angle 35°; (**b**) erosion angle 90°.

**Figure 8 materials-16-05733-f008:**
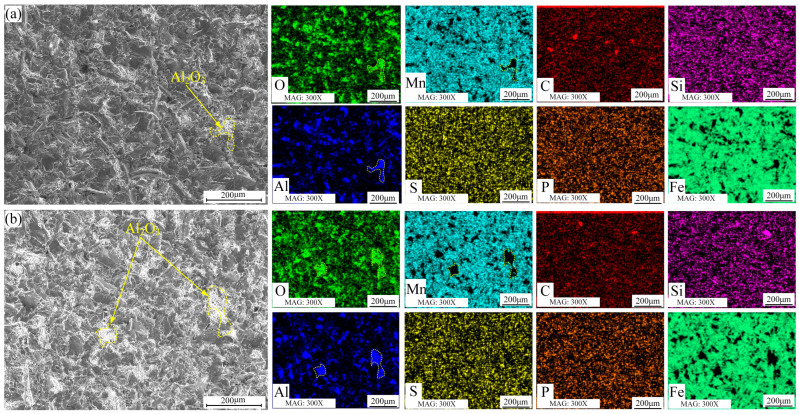
Surface morphology and element distribution of HMnS coating after erosion; (**a**) Erosion angle 50°; (**b**) Erosion angle 90°.

**Figure 9 materials-16-05733-f009:**
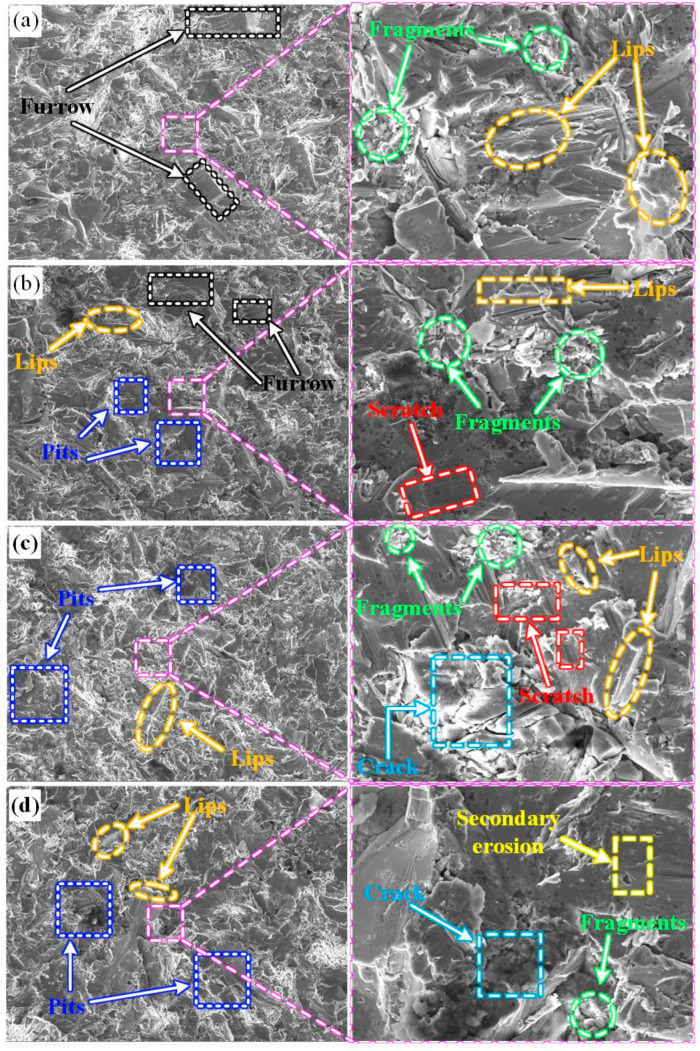
Erosion morphology at different erosion angles (**a**) 35° for HMnS plate; (**b**) 50° for HMnS coating; (**c**) 90° for HMnS plate; (**d**) 90° for HMnS coating.

**Figure 10 materials-16-05733-f010:**
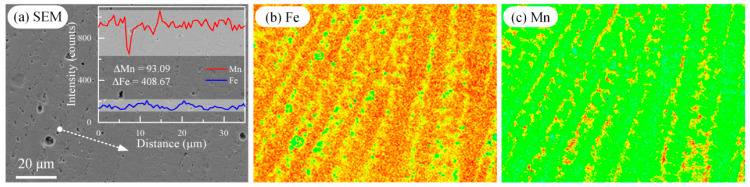
EPMA images of the cross-section of HMnS coating. (**a**) SEM image and the element line distribution of Fe and Mn at the dashed line with arrow; (**b**,**c**) represent the distributions of Fe and Mn elements corresponding to Figure (**a**), respectively.

**Figure 11 materials-16-05733-f011:**
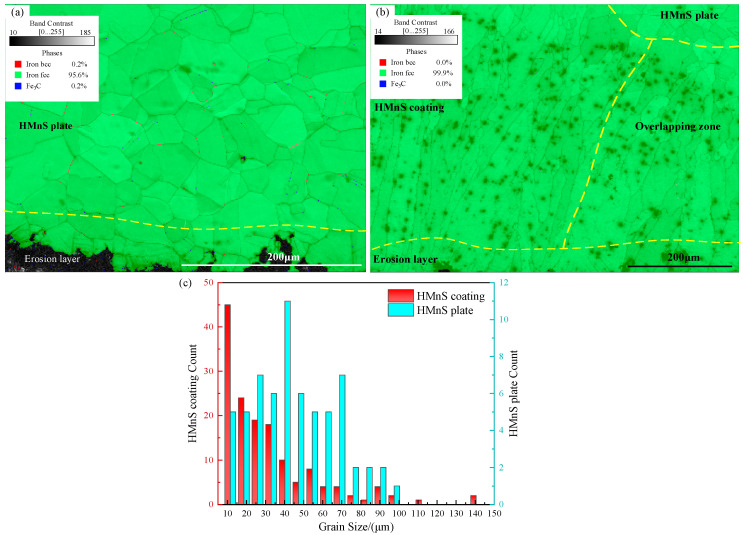
Statistical diagram of phase composition and grain size after erosion; (**a**) The phase composition of HMnS plate; (**b**) Phase composition of HMnS coating; (**c**) Statistical diagram of grain size.

**Figure 12 materials-16-05733-f012:**
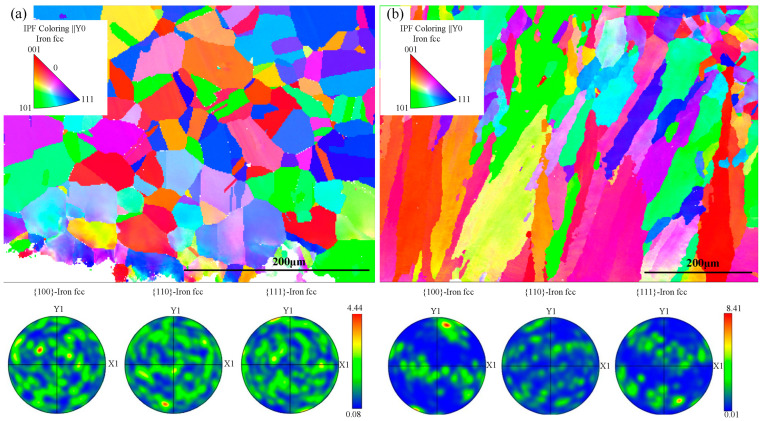
Reverse pole diagram and pole diagrams far from the erosion layer; (**a**) HMnS plate; (**b**) HMnS coating.

**Figure 13 materials-16-05733-f013:**
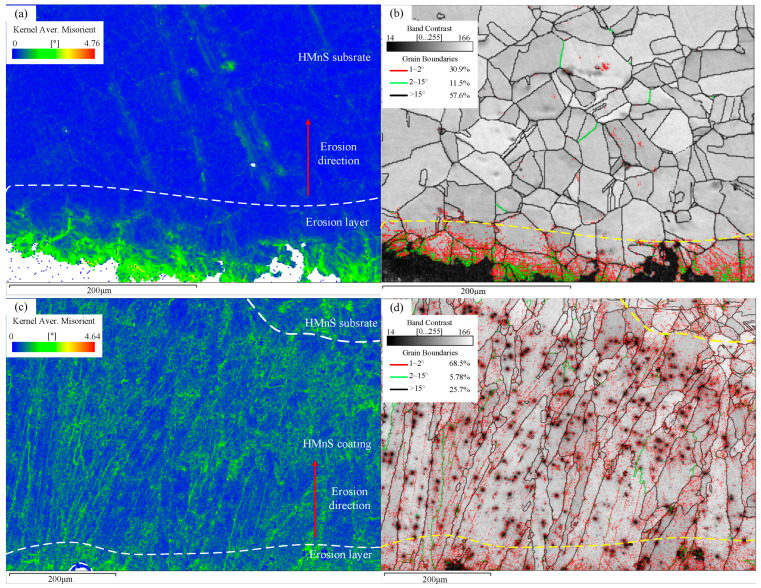
KAM and GB diagrams; (**a**,**b**) KAM and GB diagram of HMnS plate; (**c**,**d**) KAM and GB diagram of HMnS coating.

**Figure 14 materials-16-05733-f014:**
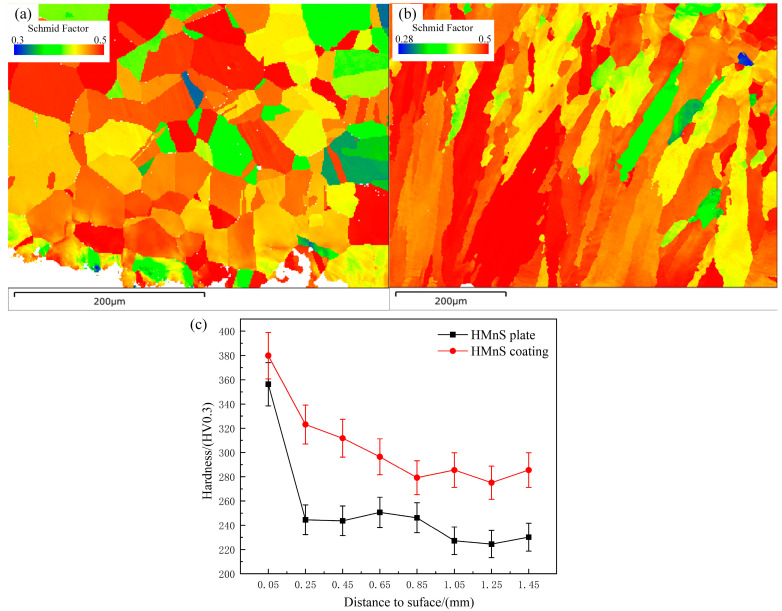
Schmidt factor distribution and section hardness distribution after erosion; (**a**) Schmidt factor distribution of HMnS plate; (**b**) Schmidt factor distribution of HMnS coating; (**c**) Cross section microhardness distribution after erosion.

**Table 1 materials-16-05733-t001:** Parameters of laser wire feeding cladding.

Wire Feeding Angle/°	Laser Power /W	Spot Diameter/mm	Scanning Speed/mm/s	Gas Flow /L/min	Overlapping Ratio/%
45	1400	2	5	15	45

**Table 2 materials-16-05733-t002:** Chemical composition of the Mn13 plate and welding wire (Atom fraction/%).

Elements	C	Mn	Si	Mo	P	S	Fe
Mn13 plate	1.13	12.84	0.47	/	0.052	0.009	Balance
Welding wire	≤1.10	11.00–18.00	0.30–1.30	≤2.50	/	/	Balance

## Data Availability

Data will be made available on request.

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
