# Peer review of "Study on Erosion Behavior of Laser Wire Feeding Cladding High-Manganese Steel Coatings"

_materials, 2023, doi:10.3390/ma16175733_

Round 1
Reviewer 1 Report
The submitted paper discusses about the erosion behavior of laser wire feeding cladding high manganese steel coatings. The researchers prepared the high manganese steel (HMnS) coating by laser wire feeding cladding technology. These researchers produced the HMnS coating using laser wire feeding cladding, which resulted in a strong metallurgical connection to the underlying material. The coating exhibited a compact microstructure without any visible cracks. At various erosion angles, the erosion rate of the coatings is lower compared to that of the substrate. The maximum erosion rate is observed at 35o and 50o, respectively. The paper has been fairly organized and can be accepted for publication.
1) In page 4, how can be it found from the diffraction peaks of the HMnS that the HMnS plate and HMnS coating were a single austenitic structure with no excessive precipitation of carbides?
2) In page 4, the authors implied that the right side of the interface were fine equiaxed dendrite grains, and the left side of the interface were columnar grains. It should be more elaborated why such a difference is observed.
3) The Figure captions ((a), (b) and (c)) are missing in Figure 4.
4) According to Fig. 5, why is the peak of erosion rate shifted to the right side with the addition of coating?
5) It should be elaborated why aluminum oxide type inclusion is observed on the erosion surface rather than other inclusions?
6) It should be explained why a large number of pits and extrusion lips appeared on the sample surfaces when the erosion angle reached 90°?
7) In page 11, why was the grain orientation of the HMnS coating relatively concentrated on the<001>direction near the erosion layer?
The English of paper is fine; however; there are some minor grammatical errors through the text. The authors should accurately read the paper and resolve the issues.
Reviewer 2 Report
This manuscript analyzes the erosion behavior of a high manganese steel (HMnS) coating manufactured by laser wire feeding cladding
The work reports the change in the microstructure with respect to a casting sample using optical microscopy, SEM, XRD, and electron backscatter diffraction (EBSD). Erosion resistance is performed at different angles using a self-made erosion testing machine. Changes in the microstructure are also analysed after the erosion test.
Although it is an interesting topic, the work is very repetitive. This becomes evident when naming "HMnS coating" 96 times, and HMnS plaque 69 times despite analyzing only 2 samples, casting and laser cladding.
The manufacturing conditions for laser cladding are fixed, the manufacturing parameters are not optimized, the manufacturing conditions are not justified, no reference to previous work is included to justify manufactured conditions. For the sample identified as "plate" it is only indicated that it is Mn13 obtained by continuous casting but the supplier is not indicated, neither for welding wire.
Another big problem that the work presents is that the results obtained are not compared with the bibliography. Only two references are mentioned in the discussion of the results, which reflects that the results have not been contrasted.
One of the references used, page 5, reference 13, is used to indicate that this type of alloy presents carbide precipitation during casting and arc welding, which causes cracks and microcrasks.
The XRD analysis indicates “no excessive precipitation of carbides” What does this statement refer to? the diffraction results are not compared with any other work. In the HMnS coating diffractogram (Figure 3. line black) a peak seems to be distinguished next to the γ-Fe (111) signal, but it is not identified. No amplification or analysis of this signal is provided. This signal seems not to be present for HMnS plate. Is the signal due to carbides?
The metallographic analysis by means of SEM is carried out at low magnifications and does not provide information regarding carbides either.
In summary, in the introduction the work is justified since the biggest problem of HMnS is the formation of carbides, which causes brittleness. The only analysis that is done to try to identify carbides is XRD and the authors only provide a qualitative comment “no excessive precipitation of carbides was found”
Figure 6 diffraction, the presence of α-Fe is identified despite being completely overlapped with the very intense alumina peak. How can its presence be guaranteed?
All figure captions contain “results …… of HMnS plate and HMnS coating; a) HMnS plate, b) HMnS coating ” Repetitions should be avoided using “ Results of ….. for HMnS samples: a) HMnS plate, b) HMnS coating.
The erosion analysis compares all the incidence angles between both samples, which makes the discussion very repetitive, without providing important information. The discussion should be more general. The same occurs with the analysis of the EBSD results.
Reviewer 3 Report
The work entitled "Study on Erosion Behavior of Laser Wire Feeding Cladding 2 High Manganese Steel Coatings" sperimentally investigates the microstructure of HMnS coating applied by laser wire feeding cladding on Mn13 plates and its resistance to erosion.
The work is well-presented and interesting; however, some point should be improved:
1. Introdution is too short; moreover, abbreviations as "HMnS" should be defined also in the introduction section;
2. HB300-500 is not clear: for a better understanding I suggest to modify to "300-500 HB";
3. The choise of process parameters presented Tab.1 should be justified; moreover, I suggest to merge Tab. 2 and Tab. 3 into only one table;
4. line 120: I think that "Leicia" could be a typo for Leica, please check;
5. Hardness methodology is taken from standards? If yes, it should be indicated in the text, as the adopted hardness scale;
6. At line 152 the authors state: "The right side of the interface were fine equiaxed dendrite grains, and the left side of the interface were columnar grains": It is not clear from Fig. 3;
7. Abbreviations as KAM and GB should be explicated to the reader; moreover, the tense for figures should be present (for example "Fig. 3 showed" should be "Fig. 3 shows").
Round 2
Reviewer 1 Report
The revisions are satisfactory. The paper can be accepted in the current format.
Reviewer 2 Report
The authors have adequately answered the questions raised, they have improved the wording of the manuscript. There are no longer constant enumerations and redundancies, which makes the text easy to follow and understand.
They have also improved the discussion by incorporating more references, as well as the introduction.
Minor remarks
The minimum and average values should be properly named and not use the expressions: KAMave, SFave, SFmin
Due to the small size of the images in figure 9, it is not possible to understand the discussion in section 3.3.1
